# Soil Moisture, Organic Carbon, and Nitrogen Content Prediction with Hyperspectral Data Using Regression Models

**DOI:** 10.3390/s22207998

**Published:** 2022-10-20

**Authors:** Dristi Datta, Manoranjan Paul, Manzur Murshed, Shyh Wei Teng, Leigh Schmidtke

**Affiliations:** 1School of Computing, Mathematics, and Engineering, Charles Sturt University, Bathurst, NSW 2795, Australia; 2Centre for Smart Analytics, Federation University Australia, Berwick, VIC 3806, Australia; 3Institute of Innovation, Science and Sustainability, Federation University Australia, Berwick, VIC 3806, Australia; 4Gulbali Institue, Charles Sturt University, Wagga Wagga, NSW 2650, Australia

**Keywords:** LUCAS data, band selection, machine learning, principal component analysis, k-fold cross validation

## Abstract

Soil moisture, soil organic carbon, and nitrogen content prediction are considered significant fields of study as they are directly related to plant health and food production. Direct estimation of these soil properties with traditional methods, for example, the oven-drying technique and chemical analysis, is a time and resource-consuming approach and can predict only smaller areas. With the significant development of remote sensing and hyperspectral (HS) imaging technologies, soil moisture, carbon, and nitrogen can be estimated over vast areas. This paper presents a generalized approach to predicting three different essential soil contents using a comprehensive study of various machine learning (ML) models by considering the dimensional reduction in feature spaces. In this study, we have used three popular benchmark HS datasets captured in Germany and Sweden. The efficacy of different ML algorithms is evaluated to predict soil content, and significant improvement is obtained when a specific range of bands is selected. The performance of ML models is further improved by applying principal component analysis (PCA), a dimensional reduction method that works with an unsupervised learning method. The effect of soil temperature on soil moisture prediction is evaluated in this study, and the results show that when the soil temperature is considered with the HS band, the soil moisture prediction accuracy does not improve. However, the combined effect of band selection and feature transformation using PCA significantly enhances the prediction accuracy for soil moisture, carbon, and nitrogen content. This study represents a comprehensive analysis of a wide range of established ML regression models using data preprocessing, effective band selection, and data dimension reduction and attempt to understand which feature combinations provide the best accuracy. The outcomes of several ML models are verified with validation techniques and the best- and worst-case scenarios in terms of soil content are noted. The proposed approach outperforms existing estimation techniques.

## 1. Introduction

Soil moisture (SM), soil organic carbon (SOC), and nitrogen content (NC) are the fundamental aspects of nature that provide territory to a broad scope of life forms, and are important for healthy food production [1,2,3,4]. SM contributes to plant development and deterioration, climate change, and carbon formation, and significantly controls the filtration, overflow, drought monitoring, and evaporation rates [5,6,7]. SOC enhances the water holding capacity of the soil and nutrient production for plants, leading to plant growth [8,9]. SM and SOC act to regulate water level and energy exchange rate, directly influencing plant health and the hydrosphere beneath [10]. NC develops plants’ structure, metabolism, and creation of chlorophyll, contributing to plant growth and food production [11]. These soil properties impact crop production, biodiversity, and canopy structure, leading to soil’s chemical and physical properties [12]. Therefore, SM, SOC, and NC are vital to soil elements and need continuous monitoring. SM frequently changes due to high evaporation [13,14]. On the other hand, measuring SOC and NC is essential in both agricultural fields and forest management, as it helps to maintain the carbon and nitrogen cycle [15,16].

The conventional measuring methods for SM include the thermogravimetric method, time-domain reflectometry (TDR) [17], heat flux soil moisture sensors, and microelectromechanical system (MEMS) [13]. In contrast, SOC can be measured by mass loss on ignition, the Walkley–Black method, humic matter colorimetry, automated dry combustion, etc. [18]. Total soil nitrogen is traditionally measured by the Kjeldahl digestion method and dry combustion method [19]. These methods are slow and require labor and money, and are only suitable for small areas of land. In order to measure soil properties for larger areas of land in a short time, HS remote sensing can be a promising solution. Therefore, it is essential to develop an accurate estimation method to predict SM, OC, and NC accurately and quickly.

Over the last two decades hyperspectral imaging (HSI) technology has been used in wide fields of application, as it contains a large amount of spectral and spatial data arranged in many layers. These sensors are implemented on satellites or airborne craft, and the data are updated continuously. In this way, real-time monitoring of the earth’s surface is possible and can cover a large area in a single image. Researchers are involved in analysing HSI in different fields, including vegetation [20] and water management monitoring [21], medical diagnosis [22,23], forensic medicine [24], the military sector [25], crime [26], imaging documentation [27], mineralogical mapping [28], food quality estimation [29], etc. In addition, SM, SOC, and NC estimation from HSI is considered a vital research issue, and is a focus of concentration for many researchers.

Several studies have been conducted to estimate SM from HS data. In [30], the authors proposed a self-organizing map (SOM) framework to predict SM. According to their results, the best SM prediction accuracy was obtained by SOM (96.78%) compared with support vector regressor (SVR) and random forest (RF). In another study, the authors studied the possibilities of Sentinel-2 data for estimating bare surface SM from HS data, and Random Forest (RF) regression models were developed [31]. Their developed model could predict SM with 91% prediction accuracy when using all bands of Sentinel-2 images. The prediction accuracy was further developed (96%) by considering four essential bands. Random forest (RF) and extreme learning machine (ELM) algorithms were used to estimate the SM from HS images in [32]. In [33], the authors adopted a gradient boost (GB) algorithm to estimate SM from unmanned aerial vehicle (UAV)-based hyperspectral data. They adopted four strategies to predict SM, with the optimal fractional order combined with the optimal multiband indices providing the best results (92.10% prediction accuracy). Based on the previous SM prediction literature, the effect of soil temperature on soil moisture prediction remains unknown. However, other machine learning models’ performance should be considered, and the average and worst SM prediction accuracy remain unknown.

On the other hand, the topsoil spectral information of the Land Use and Coverage Area body Survey (LUCAS) dataset provides opportunities to develop a model for soil SOC prediction [34,35,36]. Large-scale soil SOC mapping and prediction are possible due to the availability and quality of remote sensing data [37]. This creates new opportunities to estimate the SOC effectively, and farmers and policymakers can take the necessary steps to manage the fields on a large scale [38,39]. In [40], the authors proposed a partial least square regression (PLSR) model to predict SOC using LUCAS topsoil data. Another research perspective shows SOC prediction from Sentinal-2 data [37,41,42].

Furthermore, with SM and SOC, NC estimation from HS data has become a vital research issue and has become the concentration of many researchers. In [43,44,45], stepwise multiple linear regression was used to rapidly detect organic carbon, nitrogen, potassium, and phosphorus. A wavelet analysis and transformation algorithm was used in [46] to predict the nitrogen quantity in soil. In [47], the authors estimated soil nitrogen using near-infrared reflectance (NIR) spectroscopy with a back-propagation (BP) neural network. The authors of [48] demonstrated soil nitrogen prediction in subsided land using the local correlation maximization-complementary superiority (LCMCS) method. Several researchers have used partial least squares regression models to predict soil nitrogen [49,50]. In [51], the authors compared three methods, (stepwise multiple linear regression (SMLR), partial least squares regression (PLSR), and support vector machine regression (SVMR), to predict nitrogen content with visible/near-infrared spectroscopy. PLSR regression was used to predict various soil parameters (organic, inorganic, total carbon, CEC, pH, texture, moisture), including nitrogen, in [52].

Although satisfactory research achievements have been seen in predicting SM, SOC, and NC using HS remote sensing technology, a comprehensive study of different machine learning models is needed in order to develop a generalized approach to predict different soil contents with the help of dimensionality reduction. A number of studies have used PCA to predict SM [53,54,55], SOC [56,57,58], and NC [59,60,61]. However, they all used raw HS data and a small number of ML algorithms. Different types of machine learning algorithms have different strengths. Here, we explore a wide range of machine learning algorithms in order to determine whether their performance is better or worse than indicated in the current literature. Furthermore, we investigate whether the particular band impacts moisture, carbon, and nitrogen prediction along with the combined effects of different machine learning strategies and feature transformation by PCA with effective band selection. Hence, all the comprehensive experiments in this paper are novel in comparison to the existing literature.

This study uses three different HS datasets to predict SM, SOC, and NC. The HS feature data used in the experiments were extracted from captured HS images of soil samples. These datasets were built from HS camera images by using the average reflection/absorbance to make a CSV file. These reflection/absorbance values were then used as different features for training and testing. The SM dataset was captured by [62], and contains 125 HS bands ranging from 454 nm to 950 nm. Two HS datasets from LUCAS containing 4200 HS bands from 400 nm to 2499.5 nm were used to predict SOC and NC,. As each band may not contribute equally to predicting SM, SOC, and NC [63], the most influential band selection is essential to improve prediction performance, minimize computational time, and decrease data dimension. Choosing effective bands eliminates negative influences, allowing soil parameters to be more accurately predicted.

On the other hand, soil temperature can be considered a good feature for predicting SM. It is comparatively more easy to measure surface soil temperature than soil moisture [64]. Figure 1 shows the plot of soil temperature and corresponding soil moisture in our considered dataset. From this figure, it can be seen that SM has an inverse relationship with soil temperature. We calculated the Pearson’s correlation coefficient between SM and soil temperature, and found that these two parameters are negatively correlated (−0.79). Therefore, soil temperature has a noticeable impact on SM.

Additionally, raw data or original features may contribute little to predicting SM, SOC, and NC. Hence, PCA can be adopted to extract features more effectively [65]. PCA helps to find the correlation of all the input features and produce principal components independent of one another. Regression algorithms are faster with PCA-preprocessed data, as it substantially reduces the size of the dataset and eliminates variables that are less significant to decision-making [66]. PCA can transform the essential features from raw data, thereby reducing the feature space significantly and consistently helping to eliminate the over-fitting issue, which improves SM and SOC prediction performance.

To predict SM, SOC, and NC more accurately and precisely from HS data, effective band selection, feature transformation, and high-dimensional data reduction techniques (for example, PCA) should be considered. In this study, we have used two dimensional reduction techniques: the first to extract the most crucial feature bands, and the second to reduce dimensionality using PCA. We explore the combined effect of using PCA and effective band selection to determine the prediction performance of SM, SOC, and NC. Our study’s second contribution is finding a generalized approach that can predict soil content using our proposed methodology. Thus, we explore band selection, and feature transformation in order to understand the estimation performance with several established machine learning techniques.

The contributions of this paper are:Effectively selecting the best HS band to ensure the good performance of ML regressors in predicting SM, SOC, and NCEvaluating the effect of soil temperature on SM predictionUse of PCA to improve model prediction accuracyTo understand the combined effect of PCA and effective HS band features in predicting SM, SOC, and NCTo propose a generalized approach that can predict soil content more accurately and efficientlyEvaluating the accuracy of different ML models and comparing the results with the proposed method.

The rest of the paper is organized as follows. The HS remote sensing data and ground truth SM, SOC, and NC data are described in Section 2. The step-by-step workflow of this paper is provided in Section 3. Section 4 illustrates the results in predicting SM, SOC, and NC for different algorithms, and a comparative study is presented with validation and evaluation. We critically discuss the outcomes of this research in Section 5. Finally, this paper is concluded in Section 6 with the presentation of guidelines for SM, SOC, and NC prediction methodology in HS remote sensing.

## 2. Dataset

### 2.1. Soil Moisture Data

#### 2.1.1. Soil Moisture and Soil Temperature Data

For this study, we used the dataset captured in [62] during a five-day field campaign in May 2017 in the area of Karlsruhe, Germany. The dataset is freely available and open for research purposes under the license (available online: https://www.gnu.org/licenses/gpl-2.0.html (accessed on 8 May 2012)). The field study was performed on undistributed bare soil with no vegetation and clayey-silt type soil. The SM was measured using a TRIME-PICO time domain reflectometry (TDR) sensor, which can measure SM to a depth of 2 to 18 cm. However, the dataset we used listed the SM of soil to a depth of 2 cm, which is considered a subsurface SM. This SM value was considered the ground truth for our study. The changes in SM ranged from 25% to 42% (Figure 2), and the corresponding soil temperature at the same depth ranged from 25.5 ${}^{{}^{\circ}}$C to 44.5 ${}^{{}^{\circ}}$C (Figure 3).

#### 2.1.2. Soil Moisture Hyperspectral Data

The hyperspectral data were captured using a Cubert UHD 285 hyperspectral snapshot camera with a spectral range of 450 nm to 950 nm. The camera was mounted on a tripod 1.7 m in height. This camera can record 50 × 50 pixel images with 4 nm spectral resolution and 125 spectral bands. The dataset consisted of 679 high-dimensional data points, including 125 hyperspectral bands. Figure 4 shows the reflectance of the HS bands. For simplicity, we considered only four soil samples with different SM values. We considered soil samples with maximum SM and with minimum SM. This figure shows that a higher percentage of the SM value generates lower values of HS reflectance, and vice versa. The HS band reflection data were used to predict SM in our study.

### 2.2. LUCAS Topsoil Data

LUCAS was established by the Statistical Office of the European Union (EUROSTAT) in 2001 to create a pan-European database on landscape parameters relevant for agricultural and environmental coverage development and evaluation [67]. For non-commercial purposes, the LUCAS topsoil dataset is available from the European Soil Data Centre (ESDAC) website. The land survey has been performed every three years for a 2 × 2 km area of land in all European member states beginning in 2006 [68]. In 2009, an extension to the periodic LUCAS was granted to provide a regular, coherent, and harmonized topsoil database for Europe [67].

In this campaign, about 20,000 soil samples were accrued using a multi-level stratified random sampling technique to represent the proportion of different land use types in Europe [67]. Five topsoil samples (0–20 cm) were taken and blended into a composite sample for every sampling point. These samples were then analyzed for their physical, chemical, and reflectance properties using a standardized technique within the same laboratory [68].

After laboratory analysis of each sample, the absorbance from 400–2499.5 nm was recorded using a FOSS XDS Rapid Content Analyzer (FOSS NIRSystems Inc., Denmark) [34]. We recorded 4200 absorbance bands at 0.5 nm intervals. For this purpose of this study, we considered only the Swedish dataset, which consisted of 1891 soil samples. The dataset specifies the corresponding soil samples (point ID) with different properties such as clay, silt, sand content, pH, coarse fragments, SOC, NC, etc. The box plots in Figure 5 and Figure 6 show the interquartile range and the outlier ranges of SOC and NC, respectively. Figure 7 and Figure 8 represent the absorbance curves of different values of SOC and NC, respectively. The lowest, highest, and average range of SOC and NC was considered for simplicity. Figure 7 shows that the absorbance increases with increasing SOC and decreases with decreasing SOC in the HS band range from 400 nm to 1000 nm. However, this trend is not followed after 1000 nm to 2500 nm. On the other hand, the absorbance increases with increasing NC; however, when the value of NC is high (36.7 g/kg), the absorbance curve becomes different. The absorbance curve of NC is shown in Figure 8.

## 3. Methodology

The work was divided into two steps, namely, the prepossessing of HS bands and the regression models used to predict SM, SOC, and NC. Figure 9 represents the workflow of this study.

### 3.1. Data Prepossessing

This study considered four different steps for preprocessing methods in order to handle the extensive dimensionality in the HS data.

#### 3.1.1. Data Filtering and Mapping

Data cleaning, filtering, and mapping is the essential step for the LUCAS dataset, as it contains inhomogeneous data [69]. HS data and the corresponding ground truth of SOC and SNC were provided in different datasets. First, according to point ID (the unique ID of an individual soil sample), the soil sample HS data were mapped in two datasets for SOC and NC for Sweden. Certain soil samples’ ground truth data (SOC and NC) were missing. Therefore, these missing values and corresponding HS data were filtered from the dataset manually. All other features were eliminated to make the dataset more convenient for use and more simple and easy for training and testing purposes. However, the SM dataset was previously cleaned and mapped with corresponding HS data.

#### 3.1.2. Feature Scaling

After that, feature scaling was performed for the three HS datasets to standardize all the input data before training our model. The purpose of feature scaling is the mathematical transformation of features or independent variables to improve prediction performance. It is essential to perform the mathematical transformation of variables and make the input data balanced to ensure that their contributions are balanced. In this study, we considered the standard scaling method. The HS band data were scaled according to the transformation formula provided in Equation (Equation 1): (1)$$z_{score}\;=\;\left(x-\bar{u}\right)/sd,$$
where $z_{score}$ is the standard score, *x* is the training sample, and $\bar{u}$ and sd are the mean and standard deviation of the training sample, respectively.

#### 3.1.3. Hyperspectral Band Selection

In the third step, we selected an effective HS band range for three different HS datasets. For the large volume of data, band selection becomes more important because it saves computational time and effort. The most effective SM HS bands were selected by considering a small portion of the HS band and the SM prediction performance was noted for that particular portion of bands. Several experiments were performed to understand the performance of each particular band to estimate SM. We used trial and error methods, and listed the results at which particular bands were more significant in predicting SM. The best SM prediction was provided by the HS band ranging from 454 nm to 742 nm; therefore, this portion of HS was considered the most influential band, and bands ranging from 746 nm to 950 nm were eliminated.

On the other hand, due to many HS absorbance band ranges for the SOC and NC dataset, eliminating and selecting specific band ranges to obtain the best prediction accuracy became complex. Hence, the least absolute shrinkage and selection operator (Lasso) algorithm was applied to determine the significant band range [70]. In this experiment, we selected the 575.5–1062 nm, 1100 nm, 1852–1885 nm, 1945–2017.5 nm, 2053–2208 nm, and 2454–2499.5 nm HS bands. From the 4200 HS bands, we selected only 1591 bands and eliminated 2609 bands.

Similarly, for NC prediction the Lasso algorithm was applied to select the most influential bands. Of the 4200 bands, only 252 bands were selected as the most significant bands. The effective band ranges were 594–616.5, 646–675.5, 1052.5–1108.5, and 2302–2489. By applying the Lasso algorithm, 3948 bands were eliminated, significantly reducing computational cost and time.

This algorithm regularizes features by shrinking the regression coefficients and reducing a number of the less essential coefficients to 0. After shrinkage, only the non-zero components were used as a selected feature to train the model. Therefore, significant numbers of weak features were eliminated, improving model prediction performance and minimizing both bias and variance.

The Lasso algorithm works with the following cost function:(2)$$\frac{1}{2N_{training}}\sum_{i=1}^{N_{training}}{\left(y_{true}^{i}-y_{observed}^{i}\right)}^{2}+\alpha \sum_{k=1}^{n}\left|a_{k}\right|$$
where $a_{k}$ is the *k*-th feature coefficient, α is a hyperparameter, and $y_{true}$ and $y_{observed}$ are the ground truth and predicted data, respectively.

The value of the cost function increases with the higher coefficient value of a particular feature. Therefore, the main aim of the Lasso algorithm is to optimize the cost function by optimizing $|a_{k}|$. If the coefficient becomes large, this forces more coefficients to be 0.

The algorithm becomes an ordinary least squares regression when α is 0. On the other hand, when α increases, the variance decreases significantly and the bias increases. In this way, the Lasso algorithm eliminates irrelevant variables that do not contribute to prediction performance.

#### 3.1.4. Data Dimension Reduction

Finally, we considered the Principal Component Analysis (PCA) technique, which is widely used to handle nonlinear high-dimensional datasets and effectively decreases the dimensionality of data. Instead of using all HS bands, we relied only on the first seven principal components, which were able to extract almost 99.97% of features from all three datasets.

### 3.2. Regression Model

Different ML regression models were studied to predict SM, SOC, and NC from the HS data: Linear Regression (LR) [71], Random Forest (RF) [72], Decision Tree (DT) [73], Gradient Boosting (GB) [74], Support Vector Regression (SVR) [75], Self Organizing Map (SOM) [30], K-Nearest Neighbors (KNN) [76], and Artificial Neural Network (ANN) [77]. Most of the ML regression models were developed from the well-known library package scikit-learn, except for SOM which was implemented in Susi library packages. We used the SOM model that already implemented by [30]. Most of the regression models follow a supervised learning algorithm. However, the SOM framework consists of unsupervised learning followed by supervised SOM.

In order to achieve good prediction accuracy performance, all the machine learning regression models were tuned during the training process; their hyper-parameters are described in Table 1. However, LR, DT, and GB provide satisfactory performance without tuning. Therefore, we relied on the basic packages of the scikit-learn library model [78] and used the grid-search approach. After completing the tuning of all ML regression models and the training phase, the testing phase was started.

### 3.3. Evaluation Parameter

The efficacy of each ML model was evaluated by computing $R^{2}$, mean absolute error (MAE), and root mean squared error (RMSE). The $R^{2}$ measure explains the percentage of variation explained by two variables (test data and predicted data), MAE signifies the absolute difference between model prediction, i.e., the predicted output and ground truth value, and RMSE describes the standard deviation of the residuals (the difference between the model prediction and actual value). The value of $R^{2}$ ranges from 0 to 1. The closer the value is 1, the better the model describes the correlation between the actual and predicted value. For MAE and RMSE, a lower the value indicates better prediction accuracy [79].

The mathematical expression of these terms is provided below:(3)$$R^{2}\;=\;1-\frac{\sum {\left(y_{i}-\hat{y_{i}}\right)}^{2}}{\sum {\left(y_{i}-\bar{y_{i}}\right)}^{2}},$$
(4)$$MAE\;=\;\frac{1}{N_{sample}}\sum_{i=1}^{N_{sample}}\left|y_{i}-\hat{y_{i}}\right|,$$
(5)$$RMSE=\sqrt{\frac{1}{N_{sample}}\sum_{i=1}^{N_{sample}}{\left(y_{i}-\hat{y_{i}}\right)}^{2}},$$
where $y_{i}$ and $\hat{y_{i}}$ are the original and predicted value for the *i*th sample, respectively, $\bar{y_{i}}$ is the average value, and $N_{sample}$ is the number of samples.

## 4. Results of Model Evaluation and Validation

This section presents the performance and comparison of regression models to predict SM, SOC, and NC from three different HS datasets. The main aim of developing a model is to show good performance on unseen data. A good model provides accurate predictions for seen and unseen data that help to eliminate overfitting and underfitting. To address this problem, k-fold cross-validation can be used. We considered ten-fold cross-validation in this study to evaluate the model efficacy. Each time the whole dataset was divided into ten groups, nine data groups were used to train the model and the remaining data were used to test the model, as shown in Figure 10. The process was repeated ten times, with model performance listed each time for the different sets of testing data. Finally, the mean prediction accuracy (Equation (Equation 6)) was derived for each ML model. The experiment was carried out for the three HS datasets to predict SM, SOC, and NC.
(6)$$R_{avg}\;=\;\frac{1}{10}\sum_{i=1}^{10}R_{i},$$

### 4.1. Soil Moisture Prediction

In this study, the model was developed to predict SM from the HS data with eight feature combinations. The regression results are shown in Table 2. In the first case, all HS bands (AHSB) ranging from 454 nm to 950 nm were considered and the prediction performance was noted for eight different ML regressors. In this scenario, SVR performed the best ($R^{2}\;=\;95.43\%,\;MAE\;=\;0.49,$ and RMSE=0.80). After that, soil temperature was considered with AHSB, and improved results were obtained for the LR, RF, GB, and ANN regressors. The best result was noted for RF, with 92.85% prediction accuracy.

In the next step, instead of using the AHSB range, the effect of the selected bands (SB) ranging from 454 nm to 742 nm was considered to predict SM. After eliminating 52 bands we obtained satisfactory results, with heights of 94.31% accuracy for the SVR model and MAE and RMSE values of 0.56 and 0.88, respectively. The effect of soil temperature with SB was evaluated in the next step; GB performed best, with 92.91% accuracy.

In order to handle extensive the dimensionality of the data, PCA was performed with four cases. First, we considered AHSB and obtained improved prediction accuracy compared with AHSB for most of the regressor models, with the exception of DT and SVR. Then, PCA was performed for SB only; again, good prediction performance was noted. Finally, considering the soil-temperature effect, PCA was performed with AHSB and SB in the third and fourth cases, respectively. From Table 2, it can be seen that PCA has a good impact on predicting SM. Best results were obtained for KNN, with more than 93% prediction accuracy for both cases.

Finally, the average performance of each feature was calculated. It is clear that SB with PCA provides the best average prediction accuracy (91.62%) in terms of $R^{2}$.

Figure 11 shows the comparison box plot of eight different ML models considering three criteria: i. AHSB, ii. PCA on SB, and iii. PCA analysis of SB, including soil temperature. This figure is drawn considering ten-fold cross-validation, and for each iteration the results indicate the best, worst, mean, and median performance of each ML model for predicting SM. Therefore, this box plot reflects the results of ten-fold cross-validation used to validate our model. The circle and cross-line on the box show the mean and median, respectively. When PCA analysis of SB is considered with and without the effect of temperature, the SM prediction accuracy improves, and the best and worst prediction ranges become shorter compared with AHSB. This prediction improvement is noted for all the ML regressions we considered. SVR provides the best average prediction accuracy, with minimal fluctuation in the prediction range.

### 4.2. Soil Organic Carbon Prediction

The possibility of predicting SOC from the LUCAS dataset (Sweden) was investigated with our proposed methodology. The experiment was performed by considering four features for all the ML regressors. Table 3 represents the ten-fold cross-validation results of SOC prediction accuracy in terms of $R^{2}$, MAE, and RMSE. Four feature combinations (AHSB, SB, PCA of AHSB, and PCA of SB) were studied to explore the possibility of predicting SOC. When AHSB was considered, the RF model provided 83.98% accuracy in terms of $R^{2}$ and MAE, and the RMSE was 35.13 and 62.46, respectively. However, when only the effective bands were considered, the SVR model performed best ($R^{2}$ = 90.52%, MAE = 26.00, and RMSE = 48.36). The prediction efficiency in terms of $R^{2}$ was improved when considering PCA analysis on SB. Considering all ML models, the best average $R^{2}$, with 83.14% prediction accuracy, was obtained when PCA was performed on SB.

However, the value of MAE and RMSE was significantly large due to the inhomogeneity of the data sample [69]. It is important to consider checking the data homogeneity before beginning machine learning or statistical operations. Homogeneous data should remain in a constant trend with the changing parameters that may affect the data. However, in practice, this is almost impossible to obtain for soil data samples. From Table 3, it can be noticed that the prediction accuracy in terms of $R^{2}$ is satisfactory. However, there are high MAE and RMSE errors due to the wide variation of the SOC soil sample. Figure 12 considers AHSB and PCA of SB in order to better understand the fluctuation range of $R^{2}$ prediction, illustrating that PCA with SB shows less fluctuation and better SOC prediction.

### 4.3. Soil Nitrogen Content Prediction

The LUCAS (Sweden) dataset with soil nitrogen as a ground truth was used to understand the possibility of predicting NC from the HS dataset. Table 4 shows the prediction performance of different ML regressors with ten-fold cross-validation.

When we considered AHSB ranging from 400 nm to 2499.5 nm, SOM performed the best, with $R^{2}$ = 74.71%, MAE = 1.87, and RMSE = 3.01 prediction accuracy, whereas DT recorded the lowest value (56.60%). In the next step, the performance of the SB range was investigated to predict the best result for NC. The performance of SB was satisfactory, with the best prediction accuracy provided by ANN ($R^{2}$ = 79.05%, MAE = 1.73, and RMSE = 2.74).

The next step recorded improved prediction performance when PCA was performed with AHSB and SB. The average $R^{2}$ results show that PCA with SB provides the best NC prediction accuracy (75.69%).

Figure 13 shows the comparison box-plot between AHSB and PCA (SB-Lasso), illustrating each ML model’s best, worst, mean, and median prediction performance. The figure is drawn based on the outcomes of ten-fold cross-validation to predict NC. This figure shows that PCA with SB performs comparatively well and shortens the range between the best and worst results.

## 5. Discussion

The main aim of this paper is to determine the most effective methodology to predict SM, SOC, and NC with reasonable accuracy from the HS data. During the experiment, all the ML tuning parameters, testing and training dataset ratio, and all other parameters remained constant in order to understand the influence of different features in predicting SM, SOC, and NC more accurately and precisely. The outcomes of this study (Table 2, Table 3 and Table 4), considering the average performance of eight different machine learning algorithms, show that the combined effect of PCA with selected bands provides the best prediction accuracy for SM, SOC, and NC. The best average prediction accuracy for SM, SOC, and NC in terms of $R^{2}$ is 91.62%, 83.14%, and 75.69%, respectively. Therefore, it is clear that PCA analysis on SB is the essential feature combination that provides the best prediction for the studied soil contents.

After a critical analysis of eight different ML regression models and according to their average prediction performance, the following conclusions can be drawn.

### 5.1. Soil Moisture Prediction

The HS band can be used effectively to predict SM with good prediction accuracy; when AHSB is considered, the SVR algorithm performs best ($R^{2}$ = 95.43%, MAE = 0.49, RMSE = 0.80).Although soil temperature shows good correlation with soil moisture, the average prediction in terms of $R^{2}$ performance is not further improved, being reduced from 85.47% to 81.89% when considering the soil temperature effect with AHSB.Effective band selection has a noticeable impact on SM prediction. Very similar results are recorded after eliminating a good portion of HS data, and the average prediction results improve from 85.47% to 85.61%;PCA has a significant impact on SM prediction. The best prediction accuracy is noted for the GB regressor ($R^{2}$ = 95.98%, MAE = 0.46, RMSE = 0.76) when PCA is performed on AHSB.In terms of average response, considering all ML models, PCA analysis on influential bands provides the best SM estimation accuracy in terms of $R^{2}$ (91.62%).

### 5.2. Soil Organic Carbon Prediction

The RF model provides the best prediction accuracy in terms of $R^{2}$ (83.93%) when AHSB is considered. However, the MAE and RMSE are 35.13 and 62.46, respectively, showing unpredictable accuracy due to a higher error rate.With the Lasso algorithm used to perform band selection, the prediction accuracy is improved; SVR sees the best prediction accuracy ($R^{2}$ = 90.52%, MAE = 26.00, and RMSE = 48.36).When PCA is performed on AHSB, the prediction accuracy is improved in terms of $R^{2}$. However, the MAE and RMSE is not much improved.Finally, when PCA is applied on SB the best prediction accuracy is noted for the ANN algorithm, with $R^{2}$ = 89.27%; MAE and RMSE are 28.19 and 48.53, respectively;The SOC prediction accuracy in terms of $R^{2}$ is satisfactory, as it defines the normalized difference between actual and predicted data.However, the error rate of MAE and RMSE is high, as the variation of the soil sample is high. As MAE and RMSE indicate the absolute difference between the original value and predicted value, it seems not to work any better; however, there is good correlation.

### 5.3. Soil Nitrogen Content Prediction

Soil NC can be predicted with reasonable accuracy from HS data. When AHSB range is considered, SOM provides the best prediction accuracy ($R^{2}$ = 74.71%, MAE = 1.87, RMSE = 3.01);The prediction accuracy for all of the ML regressors is further improved when effective band selection via the Lasso algorithm is considered; the average prediction accuracy improves from 70.56% to 73.37% in terms of $R^{2}$ value.PCA analysis plays a vital role in further improving prediction accuracy, with the average prediction accuracy increasing to 73.31% when PCA is applied on AHSB.The best result is obtained when PCA is performed on effective SB for the KNN regressor, with 77.80% prediction accuracy. From the value of the average result (75.69%), it can be observed that PCA on SB is the most important feature for predicting NC from HS data.

After critically analyzing the prediction performance of SM, SOC, and NC from three different HS datasets, Table 5 summarizes the best ML regression model according to the best performance accuracy in terms of $R^{2}$. The best SM prediction is obtained by GB regressors when PCA is performed on AHSB, with 95.98% prediction accuracy. On the other hand, the SVR model performs best (90.52%) for SOC when only SB is considered. The LR model predicts the best NC with PCA analysis on SB, with 79.23% prediction accuracy.

## 6. Conclusions

In this study, we have addressed the SM, SOC, and NC prediction topology from HS data considering different ML frameworks and listed performance comparisons. While the existing methods provide good results, the proposed method provides the best results. The importance of particular HS band selection on SM, SOC, and NC prediction from the three different HS datasets is evaluated. Additionally, the effect of soil temperature on SM prediction is considered. The study was conducted using the PCA dimensionality reduction technique. Significant improvement is noted for all ML algorithms when the combined effect of PCA with an effective HS selected band is used. This study proposes a generalized approach to predict soil content more accurately and efficiently. The proposed approach saves significant computational time and provides good prediction performance using the important features. In future work, we intend to understand the physical interpretation of HS bands and the behavior of satellite HS images to predict soil components using our proposed methodology.

## Figures and Tables

**Figure 1 sensors-22-07998-f001:**
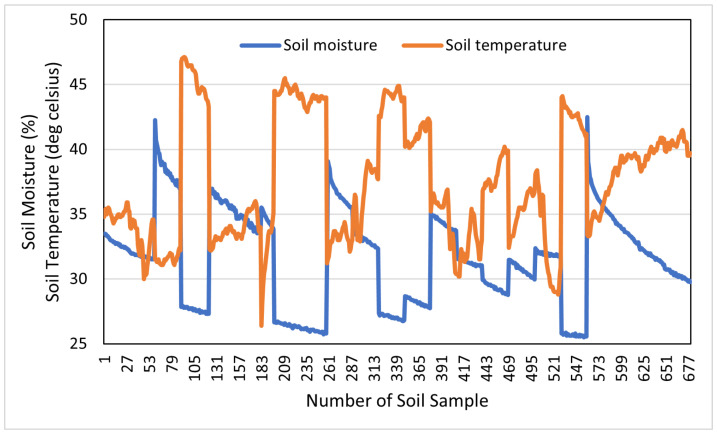
Soil moisture vs. soil temperature graph.

**Figure 2 sensors-22-07998-f002:**
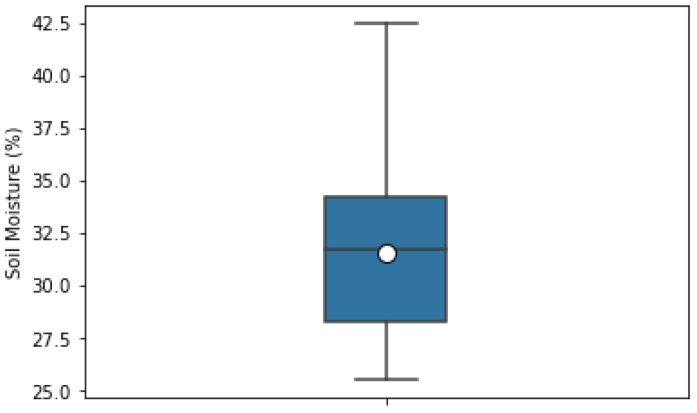
Box-plot of soil-moisture percentages, where the zero point of the box-plot represents the mean, the line intersecting the box shows the median value, and the lower and upper line of the whisker show the smallest and largest sample values.

**Figure 3 sensors-22-07998-f003:**
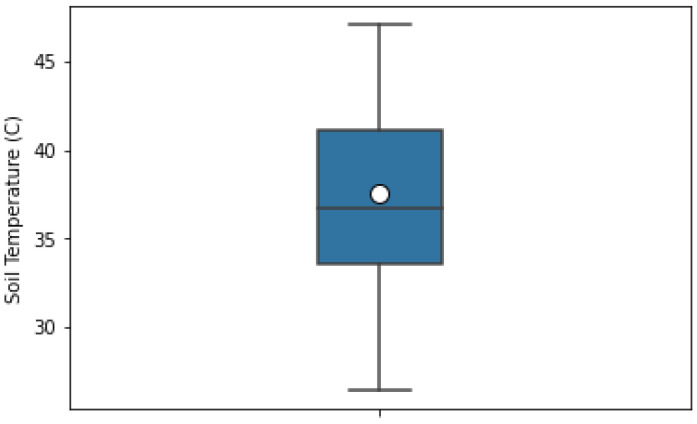
Box-plot of soil temperature interquartile range, mean, and skewness.

**Figure 4 sensors-22-07998-f004:**
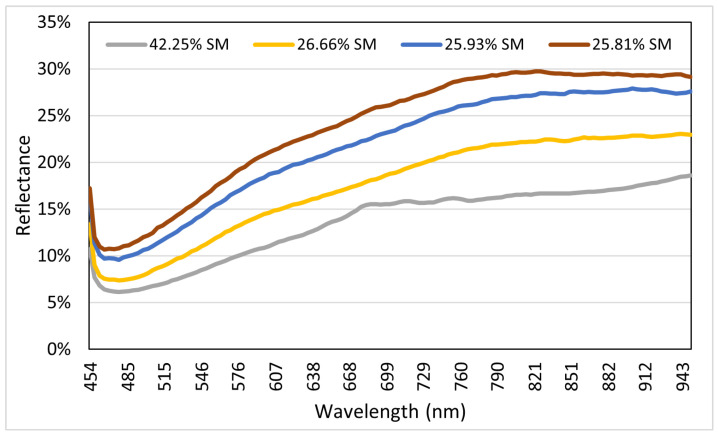
Soil moisture reflectance curve of hyperspectral band ranging from 454 nm to 950 nm.

**Figure 5 sensors-22-07998-f005:**
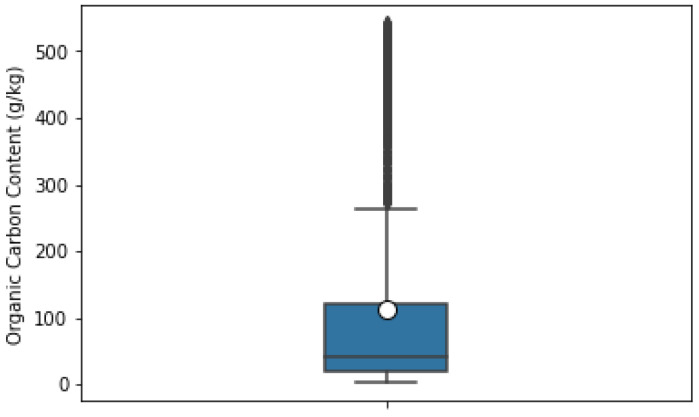
Box-plot of soil organic carbon interquartile range, mean, skewness, and outliers.

**Figure 6 sensors-22-07998-f006:**
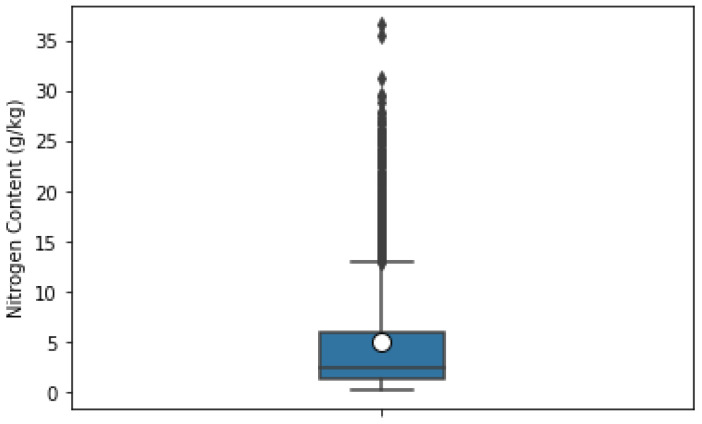
Box-plot of nitrogen content interquartile range, mean, skewness, and outliers.

**Figure 7 sensors-22-07998-f007:**
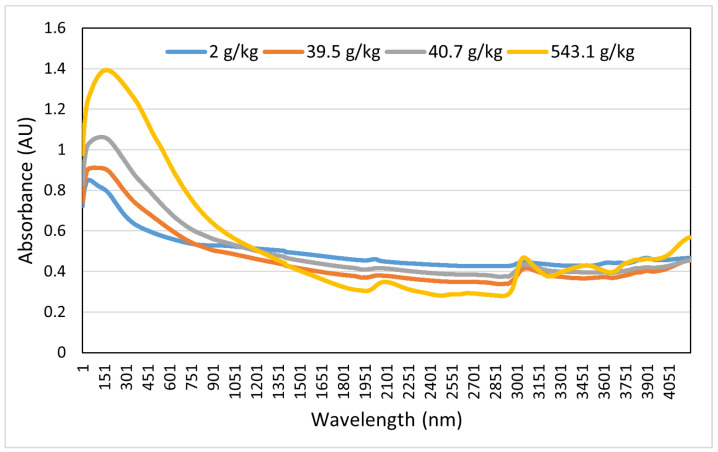
Absorbance of soil organic carbon in the hyperspectral band ranging from 400 nm to 2499.5 nm.

**Figure 8 sensors-22-07998-f008:**
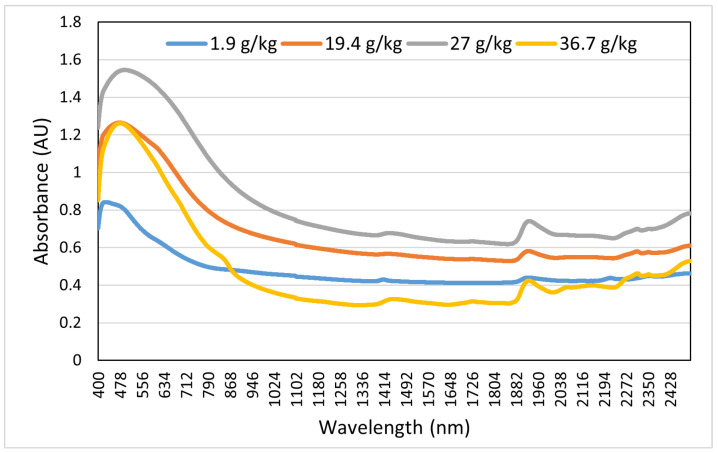
Absorbance of soil nitrogen content in the hyperspectral band ranging from 400 nm to 2499.5 nm.

**Figure 9 sensors-22-07998-f009:**
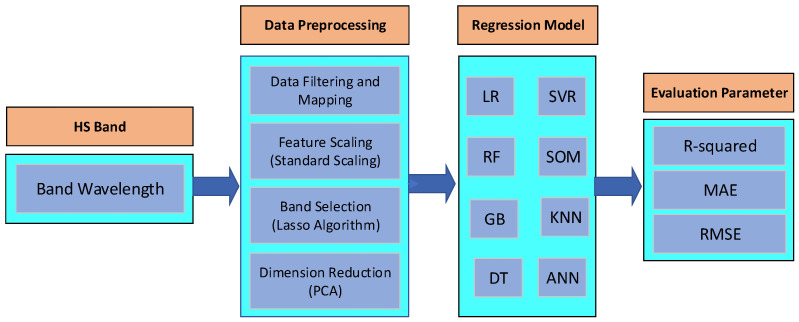
Schematic diagram of regression framework of the proposed regression investigations using different machine learning methods.

**Figure 10 sensors-22-07998-f010:**
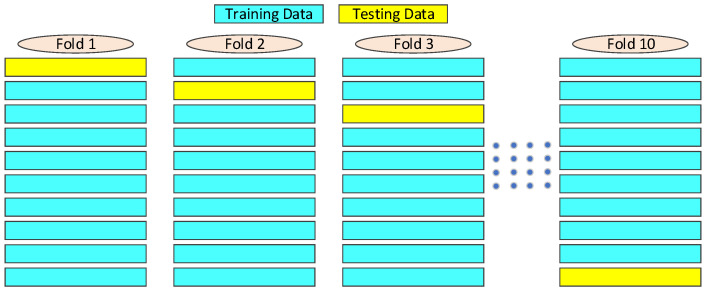
Schematic diagram of ten-fold cross validation.

**Figure 11 sensors-22-07998-f011:**
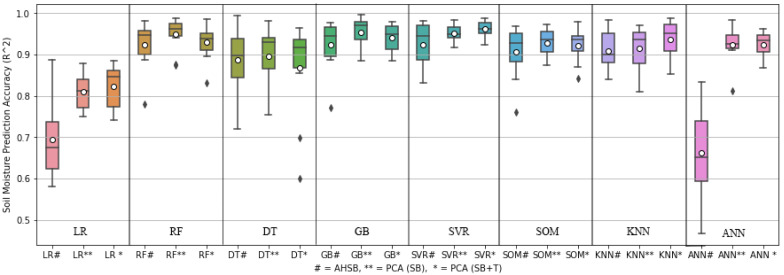
Box-plot of different ML approaches used to predict soil moisture, where black-diamond shows the outliers.

**Figure 12 sensors-22-07998-f012:**
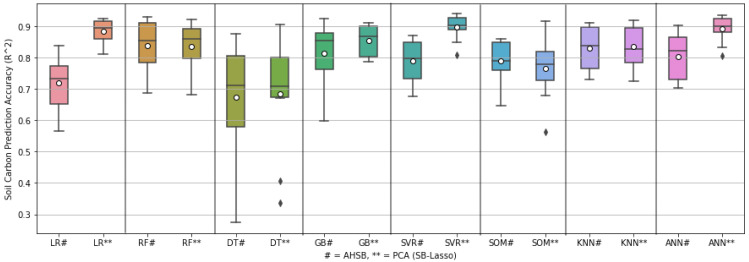
Box-plot of different ML approaches for predicting soil carbon.

**Figure 13 sensors-22-07998-f013:**
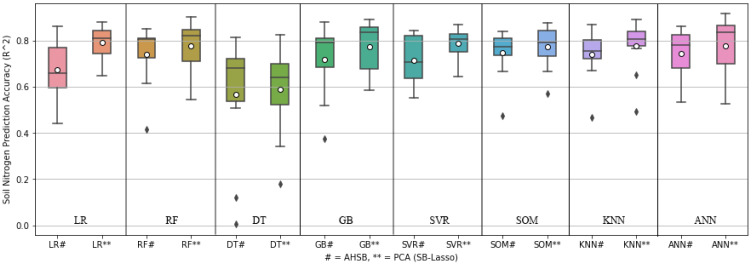
Box-plot of different ML approaches for predicting nitrogen content.

**Table 1 sensors-22-07998-t001:** Hyperparameter setup for different machine learning regression models.

Model	Library Package	Hyper-Parameter
LR	scikit-learn	–
RF	scikit-learn	bootstrap = True, criterion = squared_error, max_feature = auto, min_samples_leaf = 1, min_samples_split = 2, n_estimators = 100, n_jobs = −1
DT	scikit-learn	–
GB	scikit-learn	–
SVR	scikit-learn	*C* = np.logspace(−8, 8, 17), γ = np.logspace(−8, 8, 17), estimator = SVR(), n_iter = 30, cv = 5, n_jobs = −1, param_distributions = params, kernel = rbf, max_iter = −1, shrinking = True, tol = 0.001
SOM	susi	n_rows = 35, n_columns = 35, n_iter_unsupervised = 10,000, n_iter_supervised = 10,000, n_jobs = −1
KNN	scikit-learn	n_neighbors = 5, algorithm = auto, leaf_size = 30, metric = minkowski, weights = uniform, n_jobs = None
ANN	scikit-learn	hidden_layer_sizes = (20, 20, 20), batch_size = 10, max_iter = 500, algorithm = auto, metric = minkowski, metric_params = None, n_jobs = None, n_neighbors = 5, p = 2, weights = uniform

**Table 2 sensors-22-07998-t002:** Regression results ($R^{2}$, MAE, RMSE) for different ML models in predicting soil moisture. The bold values represent the best results (in terms of $R^{2}$) considering different ML models.

Model	AHSB ^1^	AHSB + T ^2^	SB ^3^	SB + T	PCA (AHSB)	PCA (SB)	PCA (AHSB + T)	PCA (SB + T)
LR	* 69.52%, 1.73, 2.25	69.57%, 1.72, 2.25	78.83%, 1.28, 1.68	79.18%, 1.27, 1.67	79.90%, 1.17, 1.51	81.03%, 1.16, 1.47	80.74%, 1.15, 1.49	**82.19%**, 1.13, 1.41
RF	92.35%, 0.61, 1.02	92.85%, 0.59, 1.00	92.02%, 0.61, 1.01	92.54%, 0.57, 0.99	94.73%, 0.49, 0.84	**94.99%**, 0.53, 0.89	93.71%, 0.52, 0.92	93.11%, 0.57, 0.98
DT	88.48%, 0.66, 1.34	86.15%, 0.66, 1.32	**90.93%**, 0.61, 1.22	84.68%, 0.76, 1.50	86.95%, 0.63, 1.44	90.69%, 0.54, 1.08	85.67%, 0.55, 1.24	85.47%, 0.60, 1.19
GB	92.27%, 0.64, 1.09	92.69%, 0.59, 1.08	91.85%, 0.60, 1.01	92.91%, 0.56, 0.99	**95.98%**, 0.46, 0.76	95.30%, 0.49, 0.80	95.74%, 0.51, 0.85	94.08%, 0.56, 0.90
SVR	**95.43%**, 0.49, 0.80	85.78%, 0.84, 1.31	94.31%, 0.56, 0.88	87.95%, 0.78, 1.22	91.29%, 0.62, 1.01	92.91%, 0.63, 1.12	87.79%, 0.82, 1.23	93.34%, 0.56, 0.90
SOM	89.81%, 0.72, 1.10	82.90%, 0.95, 1.43	91.20%, 0.64, 0.99	83.39%, 0.94, 1.42	92.72%, 0.63, 0.98	**93.44%**, 0.61, 0.94	93.41%, 0.55, 0.89	91.87%, 0.64, 1.01
KNN	90.95%, 0.68, 1.08	73.33%, 1.25, 1.77	88.92%, 0.76, 1.16	71.41%, 1.33, 1.84	90.39%, 0.65, 1.05	91.51%, 0.65, 1.08	**93.69%**, 0.52, 0.95	93.56%, 0.49, 0.90
ANN	64.93%, 1.42, 2.07	71.87%, 1.45, 1.90	56.85%, 1.69, 2.34	72.75%, 1.46, 2.00	90.45%, 0.61, 0.93	**93.11%**, 0.73, 1.00	90.34%, 0.95, 1.39	90.99%, 0.73, 1.02
Average Result ($R^{2}$)	85.47%	81.89%	85.61%	83.10%	90.66%	**91.62**%	90.14%	90.58%

^1^ AHSB = All Hyperspectral bands (454 nm to 950 nm), ^2^ T = Temperature, ^3^ SB = Selected bands (454 nm to 742 nm). * *R*^2^ = 69.52%, *MAE* = 1.73, *RMSE* = 2.25.

**Table 3 sensors-22-07998-t003:** Regression results ($R^{2}$, MAE, RMSE) of different ML models for predicting organic carbon. The bold values represent the best results among the different ML models.

Model	AHSB ^1^	SB ^2^ (Lasso)	PCA (AHSB)	PCA (SB-Lasso)
LR	71.94%, 55.68, 75.62	77.00%, 52.40, 77.00	81.25%, 46.03, 63.30	**88.38%**, 35.10, 50.24
RF	83.93%, 35.13, 62.46	82.60%, 35.91, 64.63	83.37%, 36.60, 62.33	**83.51%**, 35.04, 60.07
DT	67.43%, 42.94, 77.95	64.12%, 47.02, 85.58	**70.74**%, 46.53, 83.43	68.62%, 49.03, 85.52
GB	81.56%, 36.08, 64.15	82.00%, 37.24, 64.95	84.20%, 35.34, 59.43	**85.57**%, 32.62, 55.37
SVR	78.94%, 48.06, 67.39	**90.52%**, 26.00, 48.36	85.11%, 33.60, 56.98	89.72%, 27.60, 47.87
SOM	78.97%, 41.02, 67.07	**79.36%**, 41.27, 70.23	78.85%, 40.60, 66.73	76.56%, 42.89, 72.32
KNN	83.08%, 35.92, 60.92	**84.25%**, 35.52, 60.41	83.31%, 37.11, 62.74	83.56%, 35.76, 61.50
ANN	80.33%, 38.58, 62.58	77.80%, 40.77, 64.60	84.81%, 35.59, 60.14	**89.27%**, 28.19, 48.53
Average Result (in terms of $R^{2}$)	78.27%	79.70%	81.45%	**83.14%**

^1^ AHSB = All Hyperspectral bands (400 nm to 2499.5 nm), ^2^ SB = Selected bands.

**Table 4 sensors-22-07998-t004:** Regression results ($R^{2}$, MAE, RMSE) of different ML models for predicting nitrogen content. The bold values represent the best results considering different ML models.

Model	AHSB ^1^	SB ^2^ (Lasso)	PCA (AHSB)	PCA (SB-Lasso)
LR	67.40%, 2.38, 3.22	73.09%, 2.50, 3.20	79.31%, 1.85, 2.65	**79.23%**, 1.83, 2.63
RF	73.98%, 1.80, 3.03	74.57%, 1.76, 2.89	76.25%, 1.78, 2.85	**77.91%**, 1.61, 2.68
DT	56.60%, 2.29, 3.89	57.51%, 2.32, 3.97	51.77%, 2.40, 4.11	**58.87%**, 2.19, 3.75
GB	71.91%, 1.80, 3.01	74.31%, 1.79, 2.96	75.07%, 1.77, 2.84	**77.60%**, 1.63, 2.71
SVR	71.64%, 2.09, 2.99	75.63%, 1.91, 3.00	**80.57%**, 1.55, 2.66	78.97%, 1.77, 2.67
SOM	74.71%, 1.87, 3.01	76.05%, 1.79, 2.88	73.93%, 1.86, 2.99	**77.49%**, 1.78, 2.81
KNN	73.96%, 1.83, 2.95	76.79%, 1.76, 2.84	74.81%, 1.84, 2.95	**77.80%**, 1.74, 2.77
ANN	74.30%, 1.80, 2.80	79.05%, 1.73, 2.74	74.82%, 1.68, 2.68	**77.68%**, 1.66, 2.68
Average Result (in terms of $R^{2}$)	70.56%	73.37%	73.31%	**75.69%**

^1^ AHSB = All Hyperspectral bands (400 nm to 2499.5 nm), ^2^ SB = Selected bands.

**Table 5 sensors-22-07998-t005:** Best prediction results with specific feature combinations and machine learning regressor models.

Soil Property	Model	Feature Combination	Best Result ($R^{2}$)
Soil Moisture	GB	PCA (AHSB)	95.98%
Organic Carbon	SVR	SB	90.52%
Nitrogen Content	LR	PCA (SB)	79.23%

## Data Availability

Not applicable.

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
