# Peer review of "Soil Moisture, Organic Carbon, and Nitrogen Content Prediction with Hyperspectral Data Using Regression Models"

_sensors, 2022, doi:10.3390/s22207998_

Round 1
Reviewer 1 Report
Soil moisture, soil organic carbon, and nitrogen content prediction are significant soil parameters as they are directly related to plant health and food production. This manuscript used hyperspectral dataset combined with various basic statistical and machine learning algorithms to discover their potential on retrieval of the above-mentioned three paramters. Findings were found through 10-fold validation scheme that reduction of dimensions of variables and band selection helps to improve the retrieval accuracy. The work is of significance and is attractive to sceintists in related communities. However, there exist many problems regarding the innovation, method, validation and discussion of this work, as shown below.
Abstract
1.The abstract lacks descriptions of important results and findings, e.g., accounting for soil temperature does not benefit the SM prediction, and band selection helps to improve the accuracy, etc.
2.What are defined as the "existing estimation techniques"? According to the manuscript, PCA and band selection lead to more accurate estimation of the three target variables, compared with the case with all spectral bands as input. However, PCA and band selection have been already existed and quite widely-used in remote sensing models of ground features. What on earth are the innovative points of this manuscript and what problem does this work address? More clarification is required.
3. line 9: "Hyperspectral" has already been abbreviated as HS before.
4. line 13: SM has not been abbreviated so it requires its full spelling here.
5. line 16: A typo "tries to try to".
Introduction
6. The literature review part lacks strong and convincing logic. More logistic, abstract and generalized organization of this part with correct citations is needed to present the necessity and innovation of this work.
7. from line 57 to line 94: Here authors only list the brief content of previous works, and did not extract useful conclusions or inspirations on this work. Make it more concise, more comprehensive and more abstract, and extract the main point that is related to this paper.
8. line 62: A typo: there-->their
9. line 73: What is LUCAS data? Its full spellings and brief introductions are required.
10. line 98: Three or two diefferent datasets? As shown from the manuscript, one is from citation [53] and another is from LUCAS.
Dataset
11. Section 2.2: The range of SOC and SNC data should be plotted like Figure 2 or at least be introduced.
12. line 182: This trend seems not to hold after 1000 nm (Fig. 5).
Methodology
13. line 216: What is the criteria and work flow of the selection ? Is it manually selected? More clarification is suggested.
14. line 228 to line 230: Are these selected bands physically interpretable? are these bands related to specific features of the studied variables (SOC and NC), i.e., absorption of the C-N bonds? Explicit physical interpretation of the selected bands helps to generally improve the model performance.
15. Section 3.2: The optimization (or tune) work flow, optimizing ranges of model parameters and the optimized parameters should be listed.
Results
16. line 291: Is the 10-fold CV method used in 70% training dataset for parameter optimization (or tune) or in the whole dataset for validation? If it is the case of the latter, what is the use of the 70%-30% devision of the whole dataset? Please clarify.
17. line 309: Details should be introduced.
18. Figure 9: It seems that the color in this figure has no means.
19. line 343: What is called the "inhomogeneity" of the data sample? How does it lead to large errors? More illustration is suggested.
20. Table 1 and 2: The highest values are suggested to be highlighted. Bold only is not clearly seen in the table.
Discussion
21. This section is more like highlights and concluding remarks, instead of discussion. The "discussion" section should better analyze the reasons, influencing factors, sensitivity or even something new of the proposed models, methos and their results, better with illustrations.
22. line 391: This sentence is not well-understood. Extensive English editing is required to correct typos and incoherencies throughout the whole manuscript.
Reviewer 2 Report
Datta et al., has clearly represented usefulness of various regression models in data extraction, refinement of raw data using statistical methods like PCA and elucidation of hyperspectral data to calculate content of soil moisture, nitrogen and organic carbon etc. The manuscript is well drafted and can be accepted for publication in its current form. Whereas this can be more informative if the following suggestion can be included.
1.The author should explain the advantage of using PCA over other un-supervised statistical techniques for data pre-processing.
2. The author should include figure of merit of using various ML techniques to identify the best one which can be used to evaluate the hyperspectral data.
Reviewer 3 Report
A comparison analysis of different ML methods is necessary for soil components prediction, however, authors focus on using hyperspectral image for this work, no map can be found in this manuscript. So there are some serious problems in this manuscript.
(1) Figure 1, what is the physical meaning of the horizontal axis? Authors should clearly explain the preconditions of figure 1, for example, season, precipitation conditions, depth, etc.
(2) Where is the location of three datasets? Authors should add some images about the True color composite images of hyperspectral images respectively and sample distribution maps of each study area.
(3) Section 3.2, Parameters of ML methods can serious affect the performance of them, authors should clearly explain the process of parameter set and parameter values of each ML methods.
(4) Where are your prediction results in section 4? Authors must add results of each method but not just accuracy comparison tables.
Round 2
Reviewer 1 Report
The authors have addressed the questions from the reviewer.
Author Response
Thank you for your comment.
Reviewer 3 Report
(1)authors should clearly claim that the test dataset are image or no-image, if all test dataset are no-image dataset, it seems that the review works (L51-84), especially the sentinel-2 related works have no relation with this study, because the sentinel-2 image is not hyperspectral image. So this part should be removed.
(2) the parameters should be tuned by training data but not tuned before training models. besides this, the parameter description is not enough, for example, the parameters for the RF, the parameters include number of trees, number of nodes, number of samples in each node, ect.
